# *Mycobacterium tuberculosis* complex drug-resistance, phylogenetics, and evolution in Nigeria: Comparison with Ghana and Cameroon

**Muhammed Rabiu Sahal**[1,2], **Gaetan Senelle**[3], **Kevin La**[1], **Tukur Wada Panda**[4], **Dalha Wada Taura**[5], **Christophe Guyeux**[3], **Emmanuelle Cambau**[1,6], **Christophe Sola**[1,2] *

**1** Université Paris-Cité, IAME, Inserm, Paris, France, **2** Université Paris-Saclay, Gif-sur-Yvette, France, **3** FEMTO-ST Institute, UMR 6174 CNRS, DISC Computer Science Department, Univ. Franche-Comté (UFC), 16 Route de Gray, 25000 Besançon, France, **4** KNCV Tuberculosis Foundation, Nigeria, **5** Bayero University, Kano, Nigeria, **6** APHP, GHU Nord site Bichat, Service de Mycobactériologie Spécialisée et de Référence, Paris, France

* christophe.sola@universite-paris-saclay.fr

**Data Availability Statement:** All data are either freely available or published. All new SRA were submitted to the NCBI and are available under

## Abstract

In this article, we provide an *in-depth* analysis on the drug-resistance phenotypic characteristics of a cohort of 325 tuberculosis and characterize by Whole Genome Sequencing 24 isolates from Nigeria belonging to L4, L5 and L6. Our results suggest an alarming rate of drug-resistance of the L4.6.2.2 *Mycobacterium tuberculosis* complex (MTBC) lineage and a high diversity of L5. We compiled these new Sequence Read Archives (SRAs) to previously published ones from available Bioprojects run in Nigeria. We performed RAxML phylogenetic reconstructions of larger samples that include public NCBI SRAs from some neighboring countries (Cameroon, Ghana). To confront phylogenetic reconstruction to metadata, we used a new proprietary database named *TB-Annotator*. We show that L5 genomes in Northern Nigeria belong to new clades as the ones described until now and allow an update of the taxonomy of L5. In addition, we describe the L4.6.2.2 lineage in Nigeria, Cameroon and Ghana. We provide computations on the likely divergence time of L4.6.2.2 and suggest a new hypothesis concerning its origin. Finally we provide a short overview on *M. bovis* diversity in Nigeria. This study constitutes a baseline knowledge on the global genomic diversity, phylogeography and phylodynamics of MTBC in Nigeria, as well as on the natural history of this largely ignored but densely populated country of Africa. These results highlight the need of sequencing additional MTBC genomes in Nigeria and more generally in West-Africa, both for public health and for academic reasons. The likelihood of replacement of L5-L6 by L4.6.2.2 isolates, leave potentially little time to gather historical knowledge informative on the ancient history of tuberculosis in West-Africa.

bioproject n° PRJNA905483 at https://www.ncbi.nlm.nih.gov/genome.

**Funding:** The study was funded by the Petroleum Technology Development Funds under Award number P6178080548984049, PTDF and Ref n° PTDF/ED/OSS/PHD/SRM/1468/19 attributed to Muhammed Rabiu Sahal for PhD. The funders had no role in study design, data collection and analysis, decision to publish or preparation of the manuscript.

**Competing interests:** The authors have declared that no competing interests exist.

## Author summary

Nigeria is a rapidly evolving and young country that still suffers from a high-burden of tuberculosis (TB), whether linked to poverty, to HIV co-infection, or to drug-resistant TB. Genomic remains poorly implemented in Nigeria, and previous studies on the genetic diversity of the pathogen agent, *Mycobacterium tuberculosis* complex (MTBC), studied by targeted genotyping methods has first shown the high prevalence of the L4.6.2, a sublineage of the Euro-American L4 lineage. It also stressed the remaining importance of L5-L6, formerly designated as *Mycobacterium africanum*, two frequent yet poorly investigated ecotypes. Previous studies also suggested that MDR-TB transmission is alarming in Nigeria. Within a scientific project whose aim is to better analyze the genetic diversity of MTBC in Northern Nigeria and to reconstruct the evolutionary history of L5, we investigate in this article, some phenotypic and genomic characteristics of MTBC isolates obtained in a cohort of 325 TB patient from Northern Nigeria. We show an alarming rate of resistance to rifampin and isoniazid, but few indications of MDR-TB transmission. We gathered and studied a global collection of 495 Whole-genome SRA from Nigeria (n = 116), Cameroon (n = 202) and Ghana (n = 175), that includes 24 new SRAs from Northern Nigeria. The distribution of the various lineages found in these three countries support Nigeria or the border between Nigeria and Cameroon, as being the potential cradle of L4.6.2. We show that the L5 lineage is more diverse than expected and deserves more studies to better understand its evolutionary history, given the current scarcity of published L5-L6 data relatively to the wealth of L2-L4 sequences in the NCBI databases.

## 2. Introduction

Human and animal Tuberculosis (TB) is caused by *Mycobacterium tuberculosis* complex (MTBC), with 9 human and animal-adapted lineages [1]. Some of these lineages are ubiquitous, while others are restricted such as L5 and L6 in Africa. The state of tuberculosis in the world remains alarming, with more than 10,6 million people that fell ill of TB cases and 1.6 million deaths in 2022 [2]. The eradication of the disease is going much slower than expected, both in terms of the too slow decrease in number of new cases and fight against antibiotic resistance emergence and transmission. This is particularly true in Africa and in Asia, and half of the high-TB-burden countries are found in sub-Saharan Africa [2]. There are also areas where we do not really know what is going on due to lack of monitoring and adequate health capacities, as it is the case in Nigeria, that hosts one of the fastest growing population in west-Africa.

Indeed, Nigeria has the second largest TB burden in Africa, and together with five other countries (India, Indonesia, China, Pakistan and South Africa) it accounts for more than 60% of global TB cases [2]. Therefore, there is an urgent need to take stock of what is known about TB in Nigeria. A first major study investigating the genetic diversity and molecular epidemiology of MTBC was conducted in 2012 in three major cities in Nigeria: Ibadan, Nnewi and Abuja [3]. Based on spoligotyping and 24 VNTR typing of 423 clinical isolates, authors have shown that the Cameroon clade, i.e. L4.6.2 in Coll's and Napier's classification [4,5], was dominant, followed by the *M. africanum* West African I, i.e. L5, and T, mainly T2, i.e. L4.6.1 lineages [3]. They also investigated drug-susceptible and MDR-TB *M. tuberculosis* isolates and estimated the recent TB transmission rate in Nigeria. This study was followed by Molina-Moya *et al.* [6], in which spoligotyping results have been produced in 36 states, from Ziehl-Neelsen stained sputum smears on more than 500 isolates. Results are in line with the previous

study, namely: an over-representation of L4.6.2 throughout the country (half of the patterns), with an over-representation even becoming a majority of L5 in the southeast; a presence, finally, of L6 in the northwest, northeast, and west [6]. However, only two sequencing projects, namely PRJNA725394, and PRJEB15857, followed these spoligotype-based studies, leading to a limited amount of only 31 genomes available for Nigeria on the NCBI [7]. The summary of the first bioproject, which focuses on southwestern Nigeria, concludes that L4.6.2 is very prevalent in this region. The second bioproject stressed that Nigeria has an emerging problem of multidrug-resistant isolates (especially rifampin), and that some L2 isolates have become XDR-TB [8]. To sum up, L4.6.2, L5 and L6 lineages from these bioprojects have not been specifically studied *in-depth*. We should also mention PRJNA604127, which focuses on the presence of *M.bovis*. Alltogether the amount of data, as compared to the bulk of the 160,000 genomes available on the NCBI, is limited for a population of more than 200 Millions inhabitants.

This justifies the launch of new WGS studies in Nigeria, in particular on L5 and L6, which could account for up to 10 to 20 percent of total TB cases in West-Africa [9]. Using WGS, we can now go much deeper into our understanding of MTBC genomic diversity and phylogenetic reconstruction using new bioinformatic pipelines. We also wish to confirm the particular distribution by lineage through other tools, and be more precise at the level of taxonomic identification through improved sub-lineage assignment: the knowledge of the phylogeny of MTBC has indeed progressed a lot in the last 10 years, since the sequencing and characterization of lineages based on Single Nucleotide Polymorphism (SNP) [1,4,5,10–12].

We can finally try to specify the evolutionary scenario proposed in the literature, from inference of molecular-clock phylogenies, dated data being now available in sufficient number.

Using a new cohort of 325 patients representing the diversity of the north of the country (as collected in the Kano region, with available simultaneous DNA availability and clinical information on half, i.e. 165 patients), the specific sequencing by Illumina of 24 isolates of this region (mainly, L5 and L6, among which fifteen L5 sequences were submitted to the NCBI [13], this article will seek to provide some answers to the following questions: do we observe the same distribution of L4 or L5 sublineages in Nigeria as in two neighboring countries, Ghana and Cameroun and can we infer some results on L4.6.2 origin? To which specific sublineages of L4 or L5 is the emergence and transmission of antimicrobial resistance associated and what do newly sequenced isolates tell us about these lineages, regardless of the country where they were obtained.

## 3. Material and methods

### 1.1 Ethics statement

The Research project Application entitled "Comparative genomics and phylodynamics of MTBC in Nigeria with an emphasis on *Mycobacterium africanum*" was submitted to the Ethical Committe of the Aminu Kano Teaching Hospital (AKTH) (AKTH/MAC/SUB/12A/P-3/VI/3067) and approved on February 9th 2021. A formal written consent for patients to participate was not suggested by the local ethical committe, since care and public health is a basic mission of AKTH. All patient data are anonymized, and not traceable by any other third part than by the AKTH in collaboration with the Northern Nigeria Regional Reference Centre for Tuberculosis.

### 1.2 Patients and isolates data collection

The data used in this study are of two types: those already existing on NCBI, and the acquisition of a new dataset on site.

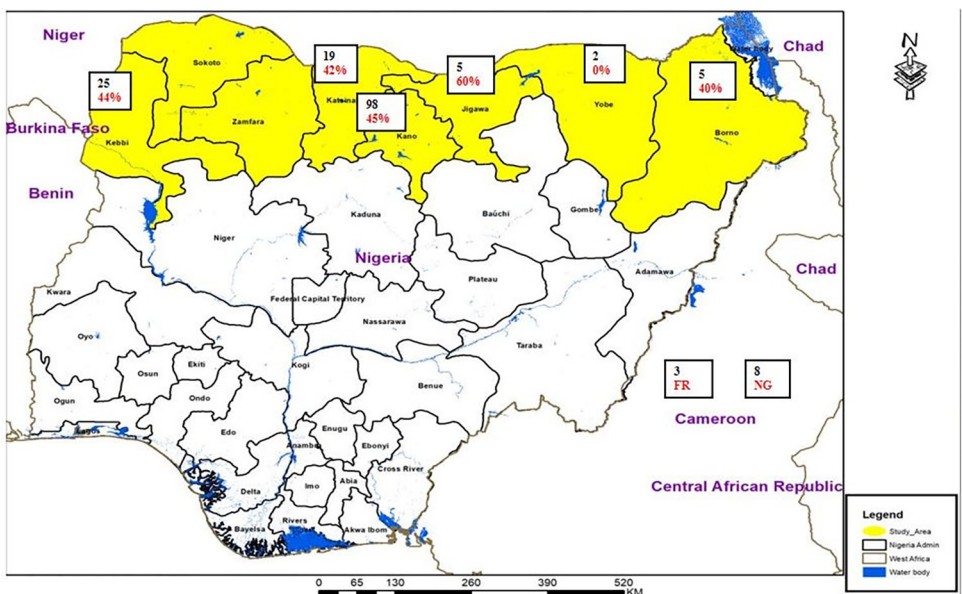

**Fig 1. Number of individuals per state in the cohort of 165 patients, in red colour, percentage of INH-resistant isolates (map source: https://www.diva-gis.org/gdata, map built using ArcGIS 10.5, https://www.esri.com/en-us).**

At the NCBI level, we investigated the references produced in the search engine of the main portal by putting the keyword "Tuberculosis" and "Nigeria". Specifically, we looked at results from Pubmed, and the bioproject and Sequence Read Archive (SRA) databases. Similar work was done on neighboring countries for comparison, replacing "Nigeria" in the query with "Benin", "Togo", "Chad", "Niger", "Cameroon", and "Ghana".

Concerning the new data, it consists on a set of 325 patients, among which a cohort of 165 patients from the AKTH (Aminu Kano Teaching Hospital) regional lab comprised complete clinical histories from six Nigerian states, namely: Borno (5), Jigawa (5), Kano (98), Katsina (19), Kebbi (25) and Yobe (2), while 8 archive DNA of Nigerian origin from two previously published studies were added and three isolates from France were also included (**Fig 1**). Patient information have been collected concerning the gender (for 157 patients), age, HIV status, and concerning the place of living: urban, semi-urban, semi-rural or rural (Table 1). At the disease level, we noted whether it was pulmonary TB or not, whether it was a new case or a retreatment case, and we also collected data on drug-resistance assessed by phenotypic drug susceptibility testing (pDST). The antibiotic susceptibility has been evaluated for the following drugs: Isoniazid (INH), Ethambutol (EMB), Rifampicin (RIF), Streptomycin (STR), Pyrazinamide (PZA), Fluoroquinolones (FQ), Amikacin (AMK), Ethionamide (ETH), Para-aminosalicylic acid (PAS), Linezolid (LZD), Kanamycin (KAN), Capreomycin (CPR), and Prothionamide (PTO). pDST was done in the Regional TB reference laboratory in Kano using a classical proportion method [14]. The question of whether this resistance could depend on geography (i.e. by local MDR-TB transmission) was further asked, and a principal component analysis (PCA) was performed by slightly noising the data (Gaussian noise, mean zero, standard deviation 0.1), to account for the fact that a resistance is never totally zero or totally complete.

### 1.3 Sputum culture and drug susceptibility test

Sputum samples were screened for drug-resistant TB by using GeneXpert (Cepheid, Sunnyvale, CA). Smear positive sputa [n = 154] cases of pulmonary TB, and/or biological specimen

**Table 1. Distribution of patient variables across the MTB Collection.**

| Variable | Category | Patient's characteristics |
|---|---|---|
| Gender | Male | 106 |
| | Female | 51 |
| | Unknown | 8 |
| Age | 18-45Years | 142 |
| | 46-65Years | 11 |
| | ≥66 years | 4 |
| | Unknown | 8 |
| HIV status | Positive | 12 |
| | Negative | 128 |
| | Unknown | 25 |
| Year of collection | 2014 | 2 |
| | 2015 | 1 |
| | 2019 | 32 |
| | 2020 | 70 |
| | 2021 | 52 |
| | Unknown | 8 |
| Case History | New | 136 |
| | Retreatment | 4 |
| | Follow-up | 12 |
| | Unknown | 13 |
| Clinical Situation | Pulmonary | 151 |
| | Extra-Pulmonary | 6 |
| | Unknown | 8 |
| Demography | Urban | 104 |
| | Semi Urban | 31 |
| | Semi Rural | 1 |
| | Rural | 16 |
| | Unknown | 13 |

for Extra-pulmonary TB cases collected from the out-patient department for TB treatment centers in AKTH Kano, (5 Northern states in Nigeria) were further processed for culture of MTBC on Löwenstein Jensen [LJ] medium at the TB regional Lab in AKTH, Nigeria with an additional 8 archive DNA samples and 3 isolates from Bichat Hospital France (Fig 1). Positive cultures (6 to 8 weeks) were examined phenotypically and the SD Bioline test, assessing the MPT64 Ag, was used for identification (Abbott, Chicago, IL). Drug susceptibility testing (DST) was done by the proportion method [14]. Once characterized, a new loopfull of organism was removed from a solid culture slope and added to 400 $\mu$l of Tris/EDTA [TE] buffer in a 1.5 ml eppendorf tube. This tube was submerged [using a lead weight] in a water bath preheated and maintained at 95°C for 30 minutes to kill mycobacteria. The tubes were sealed, covered with parafilm tape, and sent to the Associated Mycobacteriology National Reference Center in Paris-Bichat Hospital by express carrier.

## 1.4 Culture inactivation, DNA Extraction, MIRU-VNTR Screening of L5-L6, identification confirmation

We then looked more specifically for *M. africanum (Maf)* isolates, which we further sequenced by Illumina MiniSeq at the IAME-INSERM Bichat Hopital laboratory, Paris. The *Maf* isolates were initially screened based on MIRU24 copy number [15]: isolates harboring two copies of

MIRU24 were considered as *Maf* as a proxy given the known absence of L1 in this region. The experimental procedure was as follows: isolates were first heat-killed (95°C, 30 minutes in water bath) and then centrifuged for 30 minutes at 13400 rpm. 250 $\mu l$ of DNA-free water were added for initial automatized DNA extraction using a NorDiag device and kits (Nordiag group, Oslo; Norway) for 45 minutes. DNA concentrations and purity were measured on a Nanodrop spectrophotometer.

The isolates were genotyped by PCR amplification of MIRU24 using adequate primers to check for the presence of 1 copy (L2-L4) or of 2 copies (L1,L5-L6) of MIRU24. The PCR products were run on 1% agarose gel together with a 1000bp ladder, to analyze band length and infer MIRU copy number, by electrophoresis at 150V for 20 min. The reproducibility and accuracy of sizing of PCR bands, which correspond to different copy number as shown by relative migration, were checked by comparison to *M. tuberculosis* H37Rv (1 copy of MIRU24) and a reference *M. africanum* (2 copies of MIRU24). Those pre-screened isolates harboring 2 copies of MIRU24 were further subjected to a confirmatory identification test using the MTBC genotype Hain Kit (Brücker Germany, Nehren) for final identification within the MTBC.

## 1.5 Whole genome-sequencing

L5-L6 DNA was extracted again, using DNeasy UltraClean 96 Microbial Kit [Qiagen, Hilden, Germany]. A Nanodrop device [ND-ONE-W] was used to control purity and concentration for each sample [Thermofisher, Waltham, USA]. The Qiagen kit allows to obtain good quality purified DNA as assessed by a 280/260 and 260/230 ratio close to 2 and the concentration of the DNA was sufficient to perform library preparation on some samples [concentration 100 ng/$\mu l$].Samples were prepared for further sequencing. Library was produced using the Nextera XT DNA Library Preparation kit [Illumina, San Diego, USA]- with a control of the fragment size on a 2100 Bioanalyzer Instrument [Thermofisher, Waltham, USA]. Sequencing was performed using the Miniseq High Output Kit [Illumina, San Diego, USA] generating paired-end reads of 150 bp. Quality of reads was then analyzed by fastqc and a combination of web-based analysis (TB-Profiler) (available at https://tbdr.lshtm.ac.uk) or through a proprietary pipeline (*TB-Annotator*)[16].

## 1.6 Other genomic data collection

Variant calling of the fastq data has been performed using *TB-Annotator* (91,465 isolates version, [16]), after integration of the newly sequenced isolates in the database. Lineage attribution has been done by using the robust SNP barcode for typing *Mycobacterium tuberculosis* complex isolates of Coll *et al.* [4], as developed in the *TB-Annotator* platform. Sublineage information from the Coscolla *et al.* scheme [1] has been applied to isolates of Lineages 5 and 6, and it has been compared to the new proposal for L5 taxonomy published in [13]. DNA alignments of SNPs have been extracted for the following datasets:

- All L5 isolates from *TB-Annotator*, encompassing the new sequenced ones;

- All *M.bovis* isolates from *TB-Annotator*, encompassing the Nigeria ones, thanks to the selection based on the NC 000962.3:1971028:C:T SPDI (Sequence Position Deletion Insertion);

- All 4.6 (Coll's naming) that have collection dates on the one hand, and all 4.6.2.2 with collection dates on the other hand, for inference of molecular-clock phylogenies;

A phylogenetic tree has been inferred for each collection by using RAxML [17], with GTRCAT model of evolution on our supercomputer facilities. The inference of molecular-clock phylogenies has been performed by using TreeTime [18]. A maximal number of one hundred iterations have been set, with the most recent substitution model available and the

marginal maximum-likelihood assignment [19]. The outliers (whose apparent dates differ too much from the input ones) have been removed, and TreeTime has been launched ten times to investigate the stability of the produced results [18].

A potentially noisy but scalable version of the spoligotypes was obtained by aligning the reads to the 43 reference spacers with BWA[20]. Each variant indexed in the database is annotated for resistance based on TBDB data [21] and WHO data [22]. These spoligotypes and resistance extractions are done by *TB-Annotator* too. A list of mutations potentially interesting for the study of antibiotic resistance was then produced from the SNP matrix of the considered isolates (in row, the isolates, in column the mutations) and by considering for Y each resistance vector associated to each antibiotic. We then extracted the list of columns closest to the target Y in the sense of the Matthews correlation coefficient [23].

## 1.7 Bioinformatical analysis and statistics

In addition to the newly sequenced isolates and the previously mentioned bioprojects, the NCBI search recovered 48 isolates from Nigeria from the multi-country generalist project PRJEB50999 and two isolates from PRJNA825524, bringing the total to 101 isolates (plus 15 *M.bovis* from PRJNA604127). It should be noted that there is an additional three L5 isolates from Nigeria in [24], but these are PacBio data that our pipeline does not currently process. The search for isolates for neighboring countries has been relatively successful depending on the country. We found only one strain from Benin, an L5, which was not included in the *TB-Annotator* due to a quality problem. Also, surprisingly, no sequenced genomes from Togo, Chad, or Niger were available on NCBI prior to this study. However, we were able to integrate an isolate from our laboratory for Benin (since published under the accession SRR22419692), a second for Niger (SRR22419698, from the same bioproject PRJNA905483), and a final one for Togo for comparison. The PRJEB40777 project allowed us to integrate 201 representative isolates from Cameroon, and the PRJNA616081 project allowed us to do the same with 175 isolates from Ghana, see also S1 Material for further information on patients from Nigeria and S2 Material for SRAs references and origins.

Principal component analysis was performed using the PCA function of the *scikit-learn* python library. The data matrix provided had as many rows as patients in the study, and as many columns as antibiotics. The data was standardised and slightly noisy (Gaussian noise, mean zero, standard deviation 0.1), to take into account the fact that resistance is never either total or completely absent. Finally, the regions of origin were plotted using colours, using the matplotlib library.

## 1.8 Map building and map sources

Fig 1 was built using ArcGIS (version 10.5, https://www.esri.com/en-us) using copyright-free shapefiles (https://www.diva-gis.org/gdata). The second figure showing the worldwide distribution of L4.6.2.2, was built using QGIS (v3.4 Madeira), which is available from www.qgis.org and free worldwide administration boundaries shapefiles, (www.geofabrik.de).

## 4. Results and discussion

### 4.1 The patients cohort

The 165 studied patients for which DNA, clinical information and pDST were available, are distributed in characteristics described in **Table 1**. The peculiarities that can be noted can be explained in the following ways. The over-representation of women can be justified by the greater access to care for them due to maternity, while the high representation of those under

**Table 2. Phenotypic drug-susceptibility; states: BN,KB, KN, YB, JG, KT and a few samples noted NGA (unknown) or FXX (France).**

| Antibiotic | Resistance | Sensitive | Intermediate | Unknown | Total |
|---|---|---|---|---|---|
| Isoniazid (INH) | 68 | 91 | 4 | 2 | 165 |
| Ethambutol (EMB) | 4 | 150 | 4 | 7 | 165 |
| Rifampicin (RIF) | 118 | 45 | 0 | 2 | 165 |
| Streptomycin (SRM) | 14 | 138 | 5 | 8 | 165 |
| pyrazinamide (PZA) | 0 | 149 | 5 | 11 | 165 |
| fluoroquinolones (FQs) | 0 | 150 | 5 | 10 | 165 |
| amikacin (AMK) | 18 | 131 | 5 | 11 | 165 |
| ethionamide (ETH) | 0 | 149 | 5 | 11 | 165 |
| para-aminosalicylic acid (PAS) | 0 | 149 | 5 | 11 | 165 |
| linezolid (LZD) | 0 | 149 | 5 | 11 | 165 |
| kanamycin (KAN) | 18 | 131 | 5 | 11 | 165 |
| capreomycin (CPR) | 18 | 131 | 5 | 11 | 165 |
| prothionamide (PTO) | 0 | 150 | 4 | 11 | 165 |

45 years of age can be explained by the fact that Nigeria has a young population, particularly due to the birth rate and a life expectancy of 55,4 years in 2022. The vast majority of patients were HIV negative, which is unusual for TB cohorts. Similarly, the majority of cases were new, which is partly explained by the youth of the cohort. Finally, there was an over-representation of pulmonary tuberculosis cases compared to what would be expected, for reasons that remain to be explained.

We initially obtained the phenotypic drug-susceptibility profile for 165 patients from 6 Nigerian states including a few archive samples, (**Table 2** and **S1 Material**). Results turned out to present alarming rate of RIF resistance, with 70% Rifampicin resistance, and 41% Isoniazid resistance. Furthermore, the main profile within patients is a resistance to both INH and RIF, i.e. a multi-drug resistant TB (MDR-TB) pattern for 54 patients, (34.6%, cf. **S1 Material**), the second one being RIF only (44 patients, 28.2%, such a profile being reported in [25]), and the third one being the absence of resistance (21 patients, 13.4%). 39.49% of the patients are resistant to at least both INH and RIF.

The detail resistance analysis per state for 6 states is provided in **S1 Material**. Different resistance patterns are found in different states but location does not seem to be linked to the unusual drug-resistant patterns obtained in Nigeria, as shown in the Principal Component Analysis of **Fig 2**. If there are drug-resistant clusters, they are not state-dependent (no specific pattern seen in **Fig 2**), which would argue in favor of a limited MDR-TB transmission and rather of a poorly personalized treatment care.

Some of these drug-resistant isolates were further studied at the genomic level, using to search if these results could be confirmed. 14 clinical samples presenting ambiguous MIRU24 results (initially used to screen for specific WGS of L5-L6), were submitted to WGS. Eight out fourteen were either RIF monoresistant or MDR, and simultaneously shown to belong to L4.6.2.2 by WGS (72%). One L2/Beijing isolate from Yobe was even pre-XDR, the other were mixed samples (results not shown). This last WGS, experiment, even if quantitatively limited, suggests that L4.6.2.2 is associated to RIF and MDR drug-resistance in Nigeria, a result that is in line with previous published results.

## 4.2 Whole-Genome Sequencing Results of 24 isolates from Nigeria

Within the frame of a project initially intended to characterize the evolutionary genomics of L5 MTBC from Nigeria, we sequenced 24 isolates, 14 belonging to L4, 9 to L5 and one L6

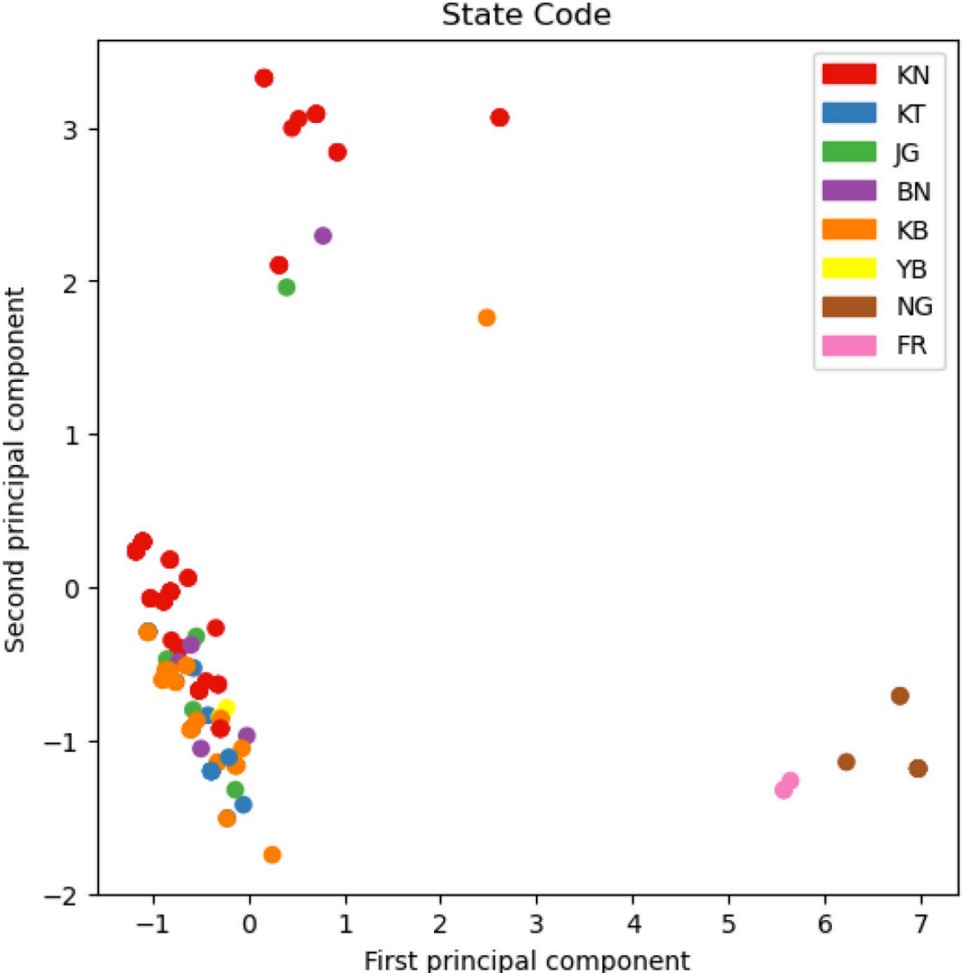

**Fig 2. Principal component analysis of drug susceptibility.** Colors are related to states, BN = Borno, KB = Kebbi, KN = Kano, YB = Yobe, JG = Jigawa, KT = Katsina, NG and FR refer to Nigeria and France samples respectively without more information (see also **S1 Material**).

sample. The results, even if quantitatively limited are in line with our expectations: L4.6.2.2 is the most prevalent clone, and is associated to multi-drug resistance; L5 is very diverse, and sample often belong to new sublineages, and shows strong regional specificity. L6 is only weakly represented and these two last lineages are less associated to multiresistance.

## 4.3 Comparison of MTBC genomic diversity and resistance in Ghana, Cameroon and Nigeria

**4.2.1 Lineage distribution.**    A detailed view per Coll's and Napier's classification, of differential MTBC genome distribution in Ghana, Cameroon and Nigeria is depicted in **Fig 3**. Lineage 4 is over-represented in this region, no matter the country, however L5 seems to be more prevalent in Nigeria and Cameroon as compared to Ghana.

There is an over-representation of Lineage 4.6 in the three countries, and more specifically of the 4.6.2.2 sublineage, as illustrated by the pie charts in **Fig 3**. In Cameroon, L5 is present but not as diverse as in Nigeria. In Nigeria, one feature is that L5.1.1 is not found (**Fig 4A**), but it is present in Ghana and Cameroon (**Fig 4B**). In general, the diversity of L5 is larger in

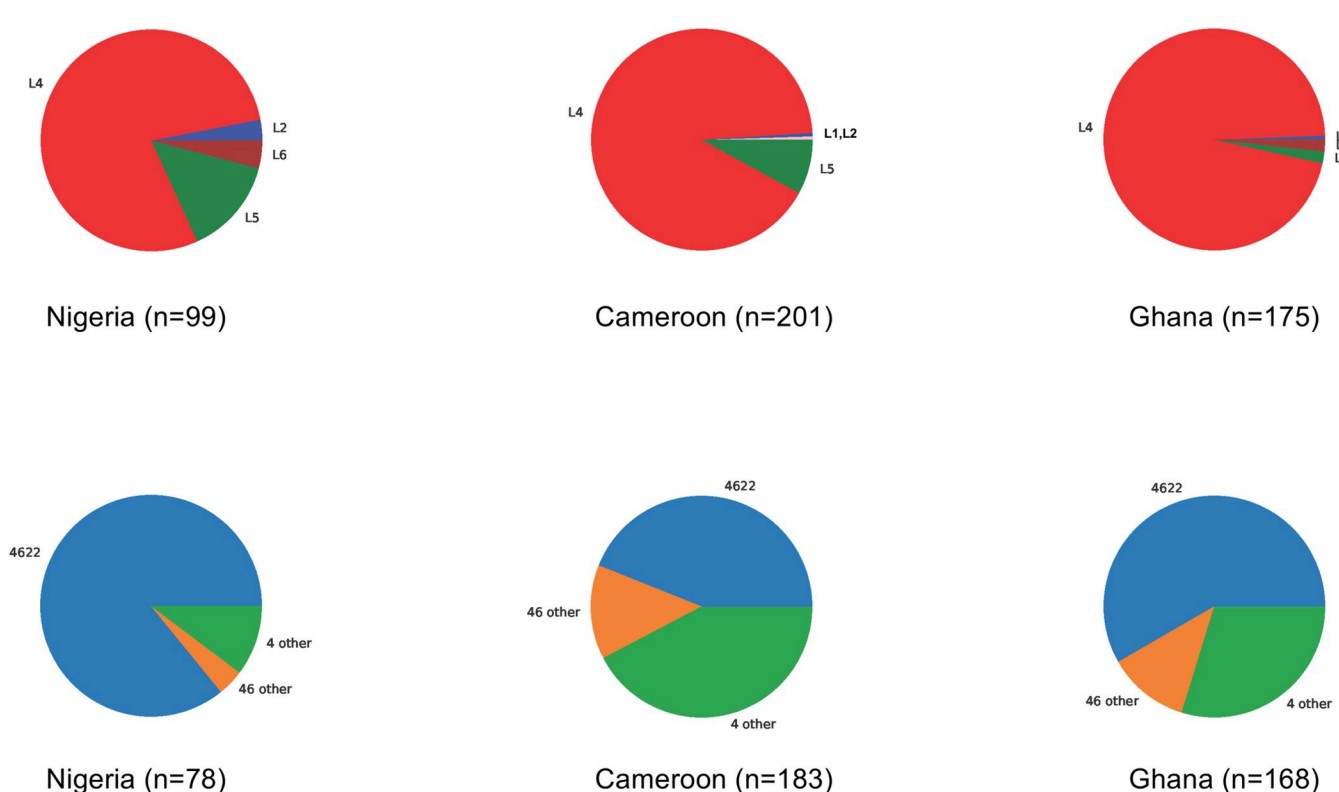

**Fig 3. Upper panel: lineage distribution per country based on WGS; lower panel: L4.6.2.2 distribution per country when compared to the other L4.**

Nigeria than in Ghana. L6 is too scarce to infer any conclusion. Another observation is that L4.1 is found in Cameroon and Ghana but not in Nigeria. In Ghana, the diversity of L4 is more important, with few L4.2 and one L4.9. In general, Lineage 2 is very rare in these three countries, and is always 2.2.1 (modern), with the noticeable exception of one 2.2.2 from Nigeria.

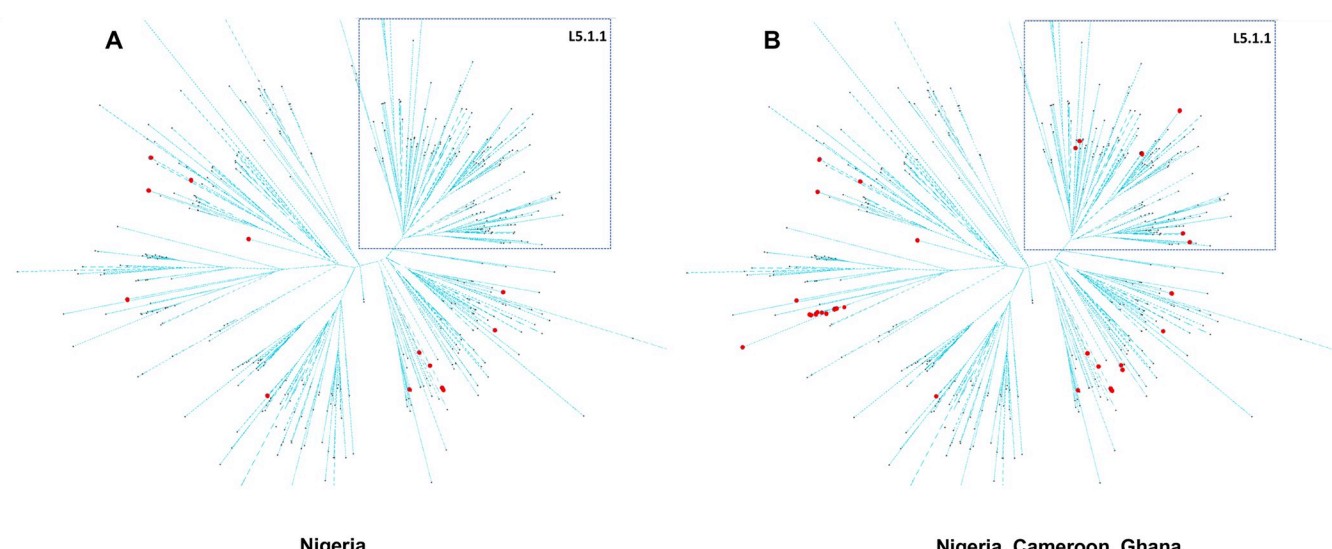

**Fig 4. Lineage 5 isolates.** L5 diversity is well represented in Nigeria, except L5.1.1 (Coscolla *et al.* naming, top right part of the network [1]).

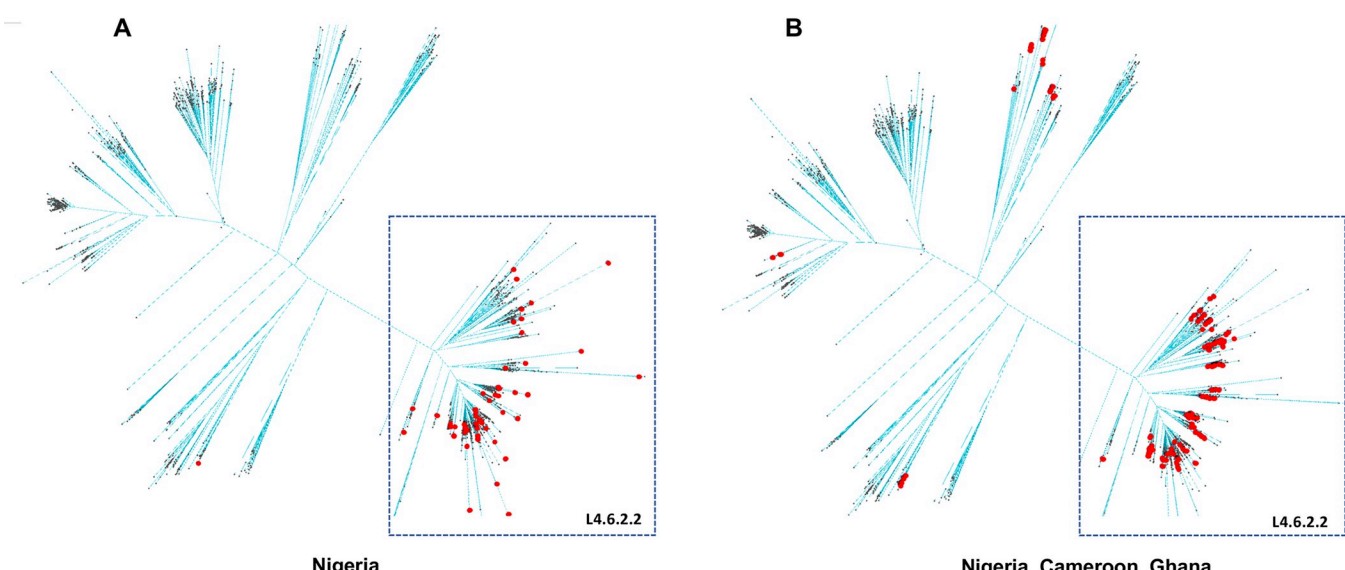

**Fig 5. Nigerian and isolates from neighboring countries within a L4.6 phylogenetic network.** Nigerian isolates are quasi-exclusively within the L4.6.2.2 clade at the bottom-right, while for other countries this is more spread in the whole network.

As shown in **Fig 5**, L4.6.2.2 is over-represented in Nigeria; 67 out of 105 non- bovis isolates, in the two major studied bioprojects and directly (8 out 14) in the last fourteen sequenced non-L5 MDR-TB isolates.

**4.2.2. Global genotypic prediction of Antibiotic resistance in Nigeria, Ghana and Cameroon.**    The global genotypic resistance profile per country, as extracted from *TB-Annotator* (>94.694 isolates) is provided in **Table 3** [16].

**Table 3. Resistance per studied country and by lineage (in percentage of isolates).**

| Country | Lineage | Total | | INH | EMB | RIF | SM | PZA | FQ |
|---|---|---|---|---|---|---|---|---|---|
| **Nigeria** | All isolates | | 99 | 36% | 26% | 45% | 28% | 13% | 0% |
| | 4 other | | 11 | 27% | 54% | 54% | 36% | 18% | 0% |
| | 4.6.2.2 | | 67 | 43% | 22% | 52% | 30% | 10% | 0% |
| **Ghana** | All isolates | | 175 | 11% | 7% | 3% | 11% | 2% | 0% |
| | 4 other | | 70 | 20% | 19% | 6% | 14% | 4% | 0% |
| | 4.6.2.2 | | 98 | 3% | 0% | 0% | 6% | 0% | 0% |
| **Cameroon** | All isolates | | 201 | 81% | 52% | 95% | 40% | 30% | 1% |
| | 4 other | | 103 | 81% | 52% | 96% | 28% | 26% | 1% |
| | 4.6.2.2 | | 80 | 80% | 50% | 96% | 56% | 25% | 0% |
| | | | | AMK | ETH | PAS | LZD | KAN | CPR |
| **Nigeria** | All isolates | | 99 | 1% | 9% | 0% | 0% | 1% | 2% |
| | 4 other | | 11 | 0% | 27% | 0% | 0% | 0% | 9% |
| | 4.6.2.2 | | 67 | 0% | 6% | 0% | 0% | 0% | 0% |
| **Ghana** | All isolates | | 175 | 0% | 7% | 0% | 0% | 0% | 6% |
| | 4 other | | 70 | 0% | 16% | 0% | 0% | 0% | 16% |
| | 4.6.2.2 | | 98 | 0% | 0% | 0% | 0% | 0% | 0% |
| **Cameroon** | All isolates | | 201 | 0% | 23% | 0% | 0% | 0% | 0% |
| | 4 other | | 103 | 0% | 27% | 0% | 0% | 0% | 0% |
| | 4.6.2.2 | | 80 | 0% | 19% | 0% | 0% | 0% | 0% |

The same trend as the one found in the 165 patients Northern Nigeria cohort, or the trend found in our sequenced isolates is observed in the genomic mutations found in *TB-Annotator*. Some differences could be explained by false positive *in vitro* tests or by false negative genomic results (unknown mutations conferring resistance). It could also be due to geographic specificities: North of Nigeria for phenotypic and 40 WGS resistance results, Southwest (PRJEB15857 bioproject), geographic origin was not available for PRJEB50999.

The resistance pattern of the Nigerian MTBC isolates essentially concerns RIF and INH: almost half of the genomes have genetic markers for RIF resistance, a third for INH. For L4.6.2.2 this pattern is specific: INH resistance decreases from 43% for the 67 L4.6.2.2 isolates to 27% for the other L4 isolates: resistance to INH thus seems to be associated specifically with L4.6.2.2. More than half of other L4 sublineages are resistant to EMB (twice as many as the rest of the collection), and to ETH (27%, compared to 9% for the rest). When comparing to the situation observed in Ghana and Cameroon, there was less drug-resistance among the 175 isolates in Ghana (10% of isolates resistant to SM and INH). In Cameroon though, the resistance is very high in the 201 genomes of our collection (95%) resistant to RIF, and more than 80% resistant to INH. More than half of the isolates are also resistant to EMB, nearly 40% are resistant to SM, 30% are resistant to PZA, and 25% to ETH. These resistances concerned 201 isolates, among which 183 belong to L4.

In 2012 Lawson *et al.* reported only 8% of MDR cases in the three cities studied [3]. However they already noted that 53% of the L4.6.2 were resistant to at least one drug, while for L5-L6 (53 isolates), one only was MDR and 32% were resistant to one of the drugs tested. They finally found a high recent transmission rate of TB (38%), indicating that the MDR-TB situation would tend to worsen, which was confirmed in between [25]. One decade later we found that around 40% of the patients are resistant to at least both INH and RIF. 52% of the L4.6.2.2 isolates are at least resistant to RIF, in line with the assessment of the drug-resistance status of the L4.6.2 lineage, that was repeatedly shown to be a problem [3,25]. Alternatively L5-L6 isolates (16 isolates over the 18 L5-6 in this study) presented less resistance. We again show that, as in [3], MDR-TB expansion is barely due to MDR-TB transmission but rather to individual lack of patient care, and is not city-dependent, but is also accompanied by a geographical gradient at a larger scale, at least along the Niger border in the case of INH.

To sum up, we observe a gradient of resistance, from Nigeria and Cameroon (more resistant) to Ghana (less resistant), with, as expected, two drugs standing out: RIF and INH. This gradient is also found within the phenotypic data for INH, in northern Nigeria (border with Niger), from west to east for unknown reasons (results not shown). In Cameroon and Nigeria, L2 isolates are even more drug-resistant. Finally, most of the studied L5 isolates from Nigeria are sensitive to antibiotics, with a few exceptions.

**4.2.3. History of the L4.6.2.2 sublineage and its phylogeography.**    All MTBC lineages can be found in Africa [26], however WGS studies remain rare for evident economic reasons in many african countries. The formerly designated"*Cameroon*" sublineage (L4.6.2.2) was originally described in Cameroon in 2003, using IS*6110*-RFLP and spoligotyping results [27]. This sublineage shows a Spoligotyping-International-Type (SIT) 61 signature, with absence of spacers 23–25 and 33–36. Variants (SIT852, SIT808, SIT403, other) were reported [28]. This lineage was also found to be prevalent, in Chad [29] (33%), Burkina Faso (30%) [30,31], Ghana (45%) [32], and Benin (35%) [33]. It was defined as L4.6.2 based on the G4125058C and the G4260742A SNPs [4,5]. Hence, L4.6.2 is the most frequent sublineage in many parts of West Africa including Nigeria [3,27,32,34]. The noose has therefore tightened to describe it as the most prevalent lineage of L4, with L4.6.2.2 (C2875883T) in Nigeria, being the most prevalent. The presence of L2 is rare in Nigeria, although not surprising, given the growing economic exchanges between West Africa and China, as shown in Benin [33]. Our Nigerian collection

contains 3 isolates belonging to the Beijing L2 lineage, two of them being studied in [8]. Another L2 pre-XDR isolate was found in Yobe.

There is a low presence of all the remaining L4s except for L4.6.2.2 and an absence of lineages 1, 3, 7, 8 and 9 in Nigeria. Even if the sample is not so large (about 100 isolates, plus 24 newly sequenced), the same result was found in several WGS studies including ours, and one might have expected to detect a greater diversity of L4, if it were really present in Nigeria. L4 prevalence was found to be 88.9% reported in for Southwest Nigeria, with a marked parallel presence of L5 and L6 [25]. Moreover, there are almost no L1 in Nigeria. Hence, L4 is poorly diverse in Nigeria, and L4.6.2.2 is the dominant clone [25]. This low L4 diversity could suggest a recent expansion of L4.6.2.2. As we will see below in the next paragraph, we will suggest some hypothesis on L4.6.2.2 emergence.

L4.6.2.2 may have emerged in the West province of Cameroon where it was first described. However, as described earlier, Nigeria is the present largest reservoir of L4.6.2.2 in Africa for both demographical and epidemiological reasons [3]. Moreover it seems to be tightly linked to the growing MDR-TB problem in Nigeria not only in Southwest but also in Northern Nigeria as shown in this work [25]. However we could not observe until now statistical significant geographical difference of prevalence of L4.6.2.2 within the 37 Nigerian states [6]. L4.6.2.2 could be over-represented in Nigeria, either thanks to an evolutionary advantage (still to be discovered) or because of an historical reason: the demographical increase in Nigeria could be associated to the emergence of this lineage. In a recent study in northern Nigeria, neither HIV prevalence nor resistance were found to be linked to the prevalence of L4.6.2.2 [35]. It has also been seen that the drug-resistance of L4.6.2.2 in Nigeria has a few remarkable features. We believe that the reasons for the epidemic success of L4.6.2.2 in Nigeria must be sought elsewhere than in antibiotic resistance.

The gradient of L4.6.2.2 in Nigeria compared to neighbouring countries, suggests that Nigeria could be its birthplace (**Fig 6**). It would have spread from its cradle, first around the Gulf of Guinea, and in relation to the growing Nigerian demographic weight. Indeed, if we look at countries further away and for which we have enough data, L4.6.2.2 is also found in Congo and in DRC (5 isolates out of 369 in TB-Profiler from DRC) but not yet in Gabon. It is also arriving in Gambia (112 isolates out of 1856 in *TB-Annotator*), and in this well WGS-characterized country, we see here again that resistance is not what helps 4.6.2.2 to take hold. It has clearly not yet arrived in Guinea (0 isolates out of 194), and has recently reached Mali (3 isolates out of 20, no resistance) and Liberia. Finally, there are 3 isolates in Sierra Leone, all drug-sensitive. However, not enough results are available in Niger and no data at all in Togo.

The "*Uganda*" and "*Cameroon*" lineages are the two respective L4.6.1 and L4.6.2 components of L4.6. Two scenarios could explain their evolution. On the one hand, an early history linked to the Bantu expansion towards southern Africa linked to agriculture development, with a cradle that could be bordering Cameroon or Nigeria, and a period of expansion that could be between 1000 BC and 500 AD [36]. On the other hand, this sublineage could be linked to an earlier human migration, and could be in Africa since up to 20,000 years ago [26], with a very slow evolution. However, the low number of SNPs in 4.6 compared to H37Rv does not strongly support this hypothesis.

The molecular-clock phylogeny of **S3 Material** places the separation between 4.6.1 and 4.6.2 between the years 1550 and 1600 A.D., a separation between 4.6.2.1 and 4.6.2.2 coming then quickly (first half of the XVIIth century), and a long sleep of 4.6.2.2 before an explosion around 1850. **Fig 6** also presents a dated phylogeny of L4.6.2.2. The reasons for the separation between 4.6 and 4.7 (Congo) on the one hand, and the appearance of the two sublineages of 4.6 on the other hand, in a handkerchief at the end of the 16th century, remain to be explained, as well as the explosion of 4.6.2.2 about 175 years ago. Several elements may have played a role.

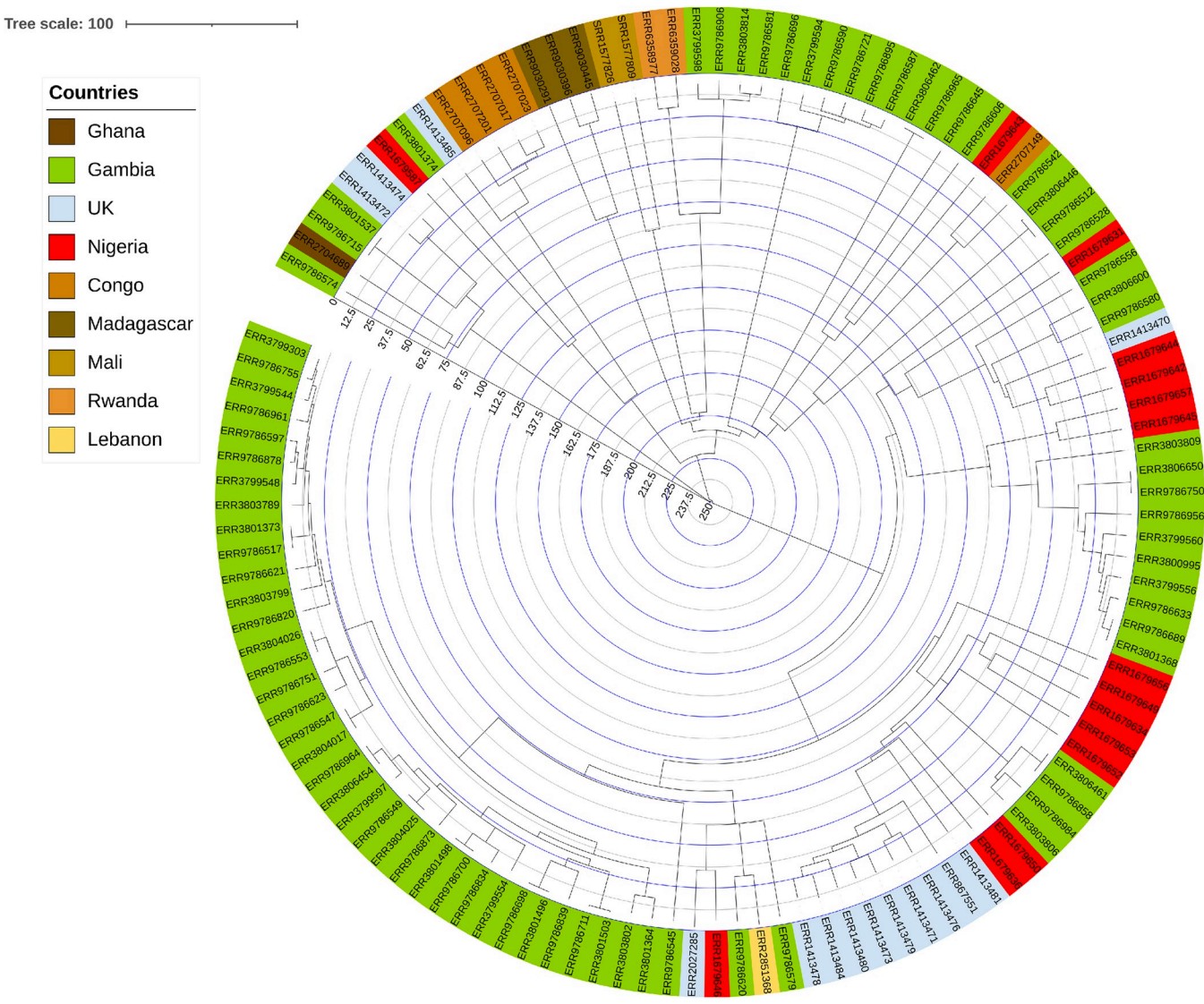

**Fig 6. Dated phylogeny of the Coll's 4.6.2.2. lineage.**

First of all, the establishment of the triangular trade. However, this migration was likely to have been preceeded by the Trans-Saharan slave trade [37]. In Nigeria, the oral tradition related to slavery is very widely associated with the activities of the kings and chieftains of the country during times preceding the event of the Trans-Atlantic slave trade. Trans-Saharan slave trade was known as early as arab written sources have been available, connecting East to West-Africa [38]. Nineteenth century movements are also likely to be linked to the history of the european colonization an in particular of the British Colonial Empire [38]. The last wave of expansion, finally, in the first half of the twentieth century, probably had much to do with the first and second world wars.

Concerning more specifically L4.6.2.2, its presence in UK could be due to the exchanges within the framework of the British colonial Empire, during the XVIIIth and XIXth century and to more recent migration events [38]. This likely contributed to the diffusion of L4.6.2.2 in Nigeria, Ghana, Benin and Gambia. This would also explain the presence of the 4.6.2.2 in

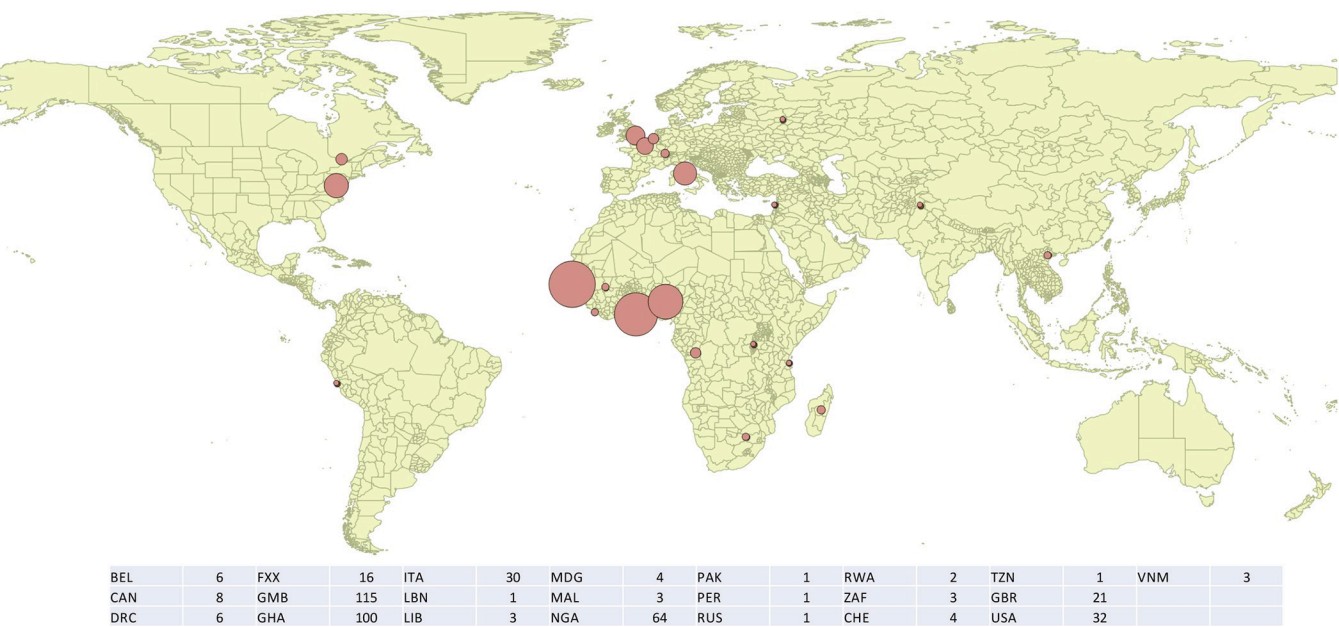

| BEL | 6 | FXX | 16 | ITA | 30 | MDG | 4 | PAK | 1 | RWA | 2 | TZN | 1 | VNM | 3 |
| CAN | 8 | GMB | 115 | LBN | 1 | MAL | 3 | PER | 1 | ZAF | 3 | GBR | 21 | | |
| DRC | 6 | GHA | 100 | LIB | 3 | NGA | 64 | RUS | 1 | CHE | 4 | USA | 32 | | |

**Fig 7. Worldwide distribution of the L4.6.2.2 isolates as recovered from *TB-Annotator* extracted data (Map built using QGIS (v3.4 Madeira), (https://www.qgis.org).**

South Africa and Tanzania, a neighboring country of Kenya, another emblematic state of the British colonial empire [38]. Indeed, the map in **Fig 7** overlaps quite well with the map of the British colonial presence on the african continent. The molecular-clock phylogeny in **S3 Material** and the one in **Fig 6** are consistent with such an explanation. It is true that the dated sample is biased with respect to the set of L4.6.2.2 isolates available online, with for example an over-representation of Gambia in a country where, as we have seen, this clade presents only 6% of the isolates. Apart from this bias due to variable sequencing from one country to another, the UK isolates may also correspond to several introduction events, going back about 75 to 100 years, i.e. to the first half of the 20th century (**Fig 6**). The two world wars may have contributed to this mixing. Thus, the war saw an important exodus of Europeans, working in administrative and commercial functions in the African colonies of the allied countries, who had to leave for the Western Front or joined units stationed in Africa to carry out campaigns outside Africa. An alternate scenario of endemicity of L4.6.2 to Western-central Africa (Cameroun-Nigeria) is also possible.

In the dated phylogeny of L4.6.2.2 (**Fig 6**), we also see that Nigeria is fairly evenly distributed in the different branches, preaching for an early establishment of this branch of TB in this country. The associated dates are all older than for the UK and Gambia, dating back to the second half of the 19th century. The other African countries, outside Nigeria, while appearing to be from older isolates, also diverged in the early 20th century, probably for the same reasons as above. Finally, the presence of L4.6.2.2 in Italy (see **Fig 7**) is undated, but can be assumed to be contemporary, stemming from the refugee crisis currently facing that country. In any case, the number of dated isolates for L4.6.2.2 is small. Its increase would allow to validate or not the proposed scenario.

A genetic geographic gradient has already been suggested, with L4.6.2 mostly identified in the north of the country, while L5 is most observed in the south and L6 is scattered [6].

**4.2.4 Phylogeography of L5 and in Western Africa.**    *M. africanum* (L5-L6) remains an important cause of tuberculosis in West Africa [39,40]. However the ecological, taxonomical,

physiological differences between L5 and L6 suggests that the very concept of *M. africanum* is outdated, much too simplistic, leading to confusion, and should not be used anymore [40]. L5 is fully diversified in our collection, in connection with the different clusters characterized previously in [3]. As such, the newly sequenced isolates described in this paper, and submitted to the NCBI, allow to deepen our understanding of the genomic diversity of Lineage5, especially concerning the discovery of rarer sublineages of L5. The names of the lineages in this description follow the thorough nomenclature of [13], with a link to [1]. We do not find L5.3, which is not surprising, considering the rarity of this lineage. In a more surprising way which may require clarification, L5.1.1 (Coscolla naming) or the new L5.1.1.2.2.3.2 is not found in Northern Nigeria even if it provides a third of the whole L5 isolates of *TB-Annotator* (see also Fig 4).

Concerning the other rare sublineages, the main gain concerns L5.1.1.2.1.2, for which there were only five known isolates before this study, and we now have ten. This doubling could will allow us to study *in-depth* this sublineage which is not epidemic but historic. It will be necessary to understand why it is present in Nigeria and still represent more than a third of the L5 isolates. The discovery of two isolates of L5.2.3.2.2 is also interesting, since nine isolates were already listed. However, their presence had so far only been reported outside Africa. Finally, the sequencing of a new isolate of a rare sublineage again illustrates the important diversity of L5 in Nigeria.

The issue of L5-L6 disappearance has been raised repeatedly in recent years, in Nigeria [6] as in Benin [33] or in Cameroon: the prevalence of L5-L6 would have dropped in 2004–2005 to 3.3% [41] in this country, while a more recent study performed in 2009 on 509 patients in the Adamaoua region reports 2.3% of these isolates [42]. It is true that the debate remains open. For instance, with 20% of L5-L6, Ghana does not seem to face such a situation [43], while some authors have suggested the possibility of culture-based counter-selection and sampling artefact as found in Burkina-Faso [31,34]. The results presented in this paper, tend to suggest a real disappearance of L5-L6. This low prevalence was also noted by authors who recently sequenced isolates in southwestern Nigeria [25], who were surprised to find a lower than expected rate (9.5%) of L5-L6 compared to the pionner historical work of Lawson *et al.*, in which the L5 alone represented 13% of the isolates [25]. In [6], authors did not detect any L5-L6 in 6 states, all in the north, central, or west part of Nigeria: L4 isolates were found instead.

The diversity of L5s found in our remote setting collection argues for an early establishment of TB due to L5 in Nigeria. We tried to date the appearance of L5 or its sublineages using Tree-Time, but it did not converge [18]. This is certainly due to the fact that we have only a few dated isolates, and that these have diverged a lot due to their age. If this is truly the case, it reinforces the demand for a stronger and specific sequencing effort of L5 in Africa: an increase of knowledge undoubtedly requires more diverse recruitment and sequencing of L5 isolates. In any case, we are leaning towards a scenario with several waves of spreading of tuberculosis in the Gulf of Guinea and West Africa: L5 and L6, in an order to be specified, followed historically by L4 and in particular L4.6, and more recently by 4.6.2.2, the latter prevailing over the others for reasons yet to be discovered. We suggest more precisely that L4.6.2.2 is taking the lead over L5 and other L4 in Nigeria, and that the replacement is fast.

To sum up, the current population structure of MTBC in Nigeria, Cameroon and Ghana reflects the specific and difficult issue to reconstitute a global and detailed Tuberculosis history in each of these countries, with many similarities, but also major differences.

**4.2.5 Last but not least: what status for *M. bovis* in Nigeria?.**   One issue that has not been investigated enough is the particular case of *M.bovis* in Nigeria (for a review see [44]). Zoonotic transmission of MTBC from human to cattle or inversely could be important in populations living close to their cattle, such as what can be seen in semi-nomadic from Northern

Nigeria and the Middle Belt with long traditions of raw milk consumption [44]. As shown in the phylogenetic tree of *M.bovis* built using *TB-Annotator* (**S4 Material**), the Nigerian *M.bovis* not only forms a separate clade, but their branches are all longer than the rest of the network, indicating a strong and separate dynamic of these Nigerian MTBC isolates since 350 years. This remarkable evolution, if confirmed, is notably illustrated in the dated tree of *M. bovis* (**S5 Material**). These isolates (upper branch) were considered as outliers among the *M.bovis* network, not fitting in the evolutionary scenario (i.e. mutation rate per year) of the rest of the species. A first explanation could be a problem in the sequences of bioproject PRJNA604127, but it would be surprising if this were to affect all isolates of this project in a uniform way. Furthermore, the only one *M.bovis* strain from Nigeria outside this bioproject (ERR1679654 from PRJEB15857) is in the same clade, and with the same long branch. This tree shape at least indicates that *M.bovis* is under evolutionary pressure in Nigeria. Another explanation would be the particular context of Nigeria, which, as we have shown here, already presents an extended specificity of evolution of L4.6.2.2 and L5. There would be a redistribution, a rapid change of lineages and a very favourable terrain, combining a demographic burst and increased economic exchanges, and a high incidence of tuberculosis in a specific context where treatment can be of poor quality for various reasons (stopping treatment before the end of the course, drug shortage or drug quality problem, long journey from home to hospital, poor mean of transport, problem of training in the care centers). For all these reasons, *M.bovis* infections should also be monitored, especially since human cases of *M.bovis* infection have been reported in Nigeria [44].

## 5. Conclusion

In a recent metaanalysis, Onyedum *et al.* showed that 32% of new cases and 53% of retreatement cases were resistant to at least one anti-TB drug in Nigeria [45]. Drug resistance indeed, is a key issue of Tuberculosis control in Nigeria. One objective of this article was to add to the knowledge base on phenotypic drug-resistant TB in Northern Nigeria, and to get insight on the genomic diversity in this large and important yet underinvestigated country, and first of all, at the level of defining circulating lineages, on the one hand on L4.6.2.2 and on the other hand on L5 and L6, both in Nigeria and in comparison to other english-speaking neighboring countries, Ghana and Cameroon.

We had hoped to be able to sequence more L5-L6 isolates, but because of a trend towards the disappearance of these lineages in this region of the world and the difficulties in growing, we were only able for the time-being to sequence about twenty five L5-L6 from DNA archives and from this cohort, and fourteen L4 isolates. The new L5 isolates sequenced, belong to rare sublineages, and are described in detail elsewhere [13]. Concerning L4.6.2.2, this study confirms their specific drug-resistance status and the interest to study them further apart at a larger scale to look for epidemiological clusters in defined settings, because they are a serious concern for public health in Nigeria.

This work contributes to a better understanding of tuberculosis in Nigeria. Nigeria is a federal country that has a large and growing population, with increase in economic exchanges due to oil exploitation, historical presence of "*M.africanum*" now better designated as L5 or L6, but TB genomics remains understudied for various technical reasons (difficulty to provide differential diagnosis on the field) and political reasons. Nigeria is also highly diverse, with growing economic resources, and a with a rich history and very diverse human genetic diversity and specific agriculture development history [46–48]. We believe that there is an urgent need to increase sequencing in this country, and more generally in sub-saharian Africa. We recall, for example, that there are much too few genomes originating from african countries in

the NCBI genome databases relatively to their demographic weight. Given the public health and also the anthropological interest to find the origin of the disease it would also be interesting to target circulating MTBC lineages, as we have done here, such as L5-L6, in new ambitious Africa-centered genomic research programs, because of the low number of genomes currently available, and because the latter is evolving and presents cases of multiple antibiotic resistance.

Our study shows the great wealth and heterogeneity of sequencing data currently available (more than 160,000 *M. tuberculosis* genomes), a wealth which, in our opinion, is currently underexploited. It also emphasizes the importance of paying special attention, when submitting new data, to specify in detail the place and date of extraction of the studied isolates. Finally, it shows the interest and urgency to develop tools such as the *TB-Annotator* that are currently lacking to question such a large amount of data.

## Supporting information

**S1 Material. Cohort composition of 165 TB patient from Northern Nigeria (6 states) with available demographic and medical data and drug resistance status of the isolates.**
(XLSX)

**S2 Material. List of all NCBI gathered SRAs accession n˚, Bioprojects n˚, our own Ref IDs, and lineage assignation according to different investigators.**
(XLSX)

**S3 Material. Dated Phylogeny of L4.6 diversification with a computed date of divergence between 4.6.1 and 4.6.2.**
(PDF)

**S4 Material. Highlight (red spots) of a particular Nigerian clade within the global *M.bovis* phylogenetical network.**
(PDF)

**S5 Material. A dated phylogeny of *M. bovis*.** The Nigerian clade is at the top of the tree.
(PDF)

## Acknowledgments

We are deeply grateful to Prs Lovett Lawson†, and Luis Cuevas† without whom this study would have never existed. We would also like to thank Dr. Jose Dominguez, Dr. Barbara Molina-Moya, for giving us access to MTBC isolates and/or archival material from previous studies. We are grateful to Pr Erick Denamur, Head of IAME, UMR1137 INSERM-Université Paris-Cité for his welcome and support. We are grateful to Mustapha Gidado (Executive Director of KNCV Tuberculosis Foundation) for connecting us with M. Tukur Panda, with KNCV in Kano, Nigeria. All computations have been performed on the"Mésocentre de Franche-Comté super-computer facilities".

## Author Contributions

**Conceptualization:** Gaetan Senelle, Christophe Guyeux, Christophe Sola.

**Data curation:** Gaetan Senelle, Kevin La, Dalha Wada Taura.

**Formal analysis:** Muhammed Rabiu Sahal, Christophe Guyeux.

**Funding acquisition:** Emmanuelle Cambau, Christophe Sola.

**Investigation:** Muhammed Rabiu Sahal, Gaetan Senelle, Kevin La, Tukur Wada Panda, Dalha Wada Taura, Christophe Guyeux.

**Methodology:** Gaetan Senelle, Christophe Guyeux, Christophe Sola.

**Project administration:** Kevin La, Tukur Wada Panda, Dalha Wada Taura, Emmanuelle Cambau, Christophe Sola.

**Resources:** Tukur Wada Panda, Dalha Wada Taura.

**Software:** Muhammed Rabiu Sahal, Gaetan Senelle, Kevin La, Christophe Guyeux.

**Supervision:** Christophe Guyeux, Emmanuelle Cambau, Christophe Sola.

**Validation:** Kevin La, Tukur Wada Panda, Christophe Sola.

**Visualization:** Gaetan Senelle.

**Writing – original draft:** Christophe Guyeux, Christophe Sola.

**Writing – review & editing:** Muhammed Rabiu Sahal, Emmanuelle Cambau, Christophe Sola.

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
