## [Decision Letter · Decision Letter 0]

19 Jul 2023

Dear Dr. Sola,

Thank you very much for submitting your manuscript "Mycobacterium tuberculosis complex drug-resistance,genomics diversity, and evolution in Nigeria: comparison with Ghana and Cameroon" for consideration at PLOS Neglected Tropical Diseases. As with all papers reviewed by the journal, your manuscript was reviewed by members of the editorial board and by several independent reviewers. The reviewers appreciated the attention to an important topic. Based on the reviews, we are likely to accept this manuscript for publication, providing that you modify the manuscript according to the review recommendations. 

Sincerely,

Simon Rayner

Academic Editor

Ana LTO Nascimento

Section Editor

Reviewer's Responses to Questions

**Key Review Criteria Required for Acceptance?**

**Methods**

-Are the objectives of the study clearly articulated with a clear testable hypothesis stated?

-Is the study design appropriate to address the stated objectives?

-Is the population clearly described and appropriate for the hypothesis being tested?

-Is the sample size sufficient to ensure adequate power to address the hypothesis being tested?

-Were correct statistical analysis used to support conclusions?

-Are there concerns about ethical or regulatory requirements being met?

Reviewer #1: Please make clearer in which part(s) the 325 patients were used and in which the 165 patients "Concerning the new data, it consists on a set of 325 patients, among which 165

samples out of this patient cohort, were collected from TB regional lab in Aminu Kano

Teaching Hospital [AKTH] from Northern States between 2019-2022, (Figure 1). "

The methods are appropriate for the objective, the link between the data and the conclusion must made clearer.

The statistics used are not clearly stated.

Reviewer #2: Are the objectives of the study clearly articulated with a clear testable hypothesis stated?

While this study does not have a specific hypothesis stated, it aims to address several questions regarding the distribution of L4 and L5 sublineages in Nigeria compared to neighboring countries, such as Ghana and Cameroon, and to infer information about the origin of L4.6.2. It also seeks to investigate the association between specific sublineages of L4 and L5 and the emergence and transmission of antimicrobial resistance.

-Is the study design appropriate to address the stated objectives?

Yes

-Is the population clearly described and appropriate for the hypothesis being tested?

The authors utilized genomic data from Nigeria (n=116), Cameroon (n=202), and Ghana (n=175), along with an additional analysis of 24 strains sequenced from Northern Nigeria.

-Is the sample size sufficient to ensure adequate power to address the hypothesis being tested?

The authors acknowledge the importance of analyzing tuberculosis strains in Northern Nigeria; however, it is noted that the number of strains used in the analysis is limited and may not be considered sufficient.

-Were correct statistical analysis used to support conclusions?

Yes

-Are there concerns about ethical or regulatory requirements being met?

No

**Results**

-Does the analysis presented match the analysis plan?

-Are the results clearly and completely presented?

-Are the figures (Tables, Images) of sufficient quality for clarity?

Reviewer #1: The supplemental data must be presented with more context.

The "supercomputer analysis" is too much a black box and must be further explained.

Reviewer #2: -Does the analysis presented match the analysis plan?

Yes

-Are the results clearly and completely presented?

The numbers representing resistance rates in Figure 6 lack clarity regarding the underlying population.

-Are the figures (Tables, Images) of sufficient quality for clarity?

The authors should re-draw the molecular clock phylogenetic analysis in Supplementary Material 5 so that the strain name can be read.

**Conclusions**

-Are the conclusions supported by the data presented?

-Are the limitations of analysis clearly described?

-Do the authors discuss how these data can be helpful to advance our understanding of the topic under study?

-Is public health relevance addressed?

Reviewer #1: The conclusion the authors suggest, is not clearly supported by the data and the link between the data and the result must be stated more clearly.

Limitations of analysis and possible alternative scenarios should be expanded upon. 

Looking at table 3 I disagree, that "This result confirms in Northern Nigeria the previous

distribution maps obtained 5 years ago by spoligotyping directly on sputum samples

[6] and definitively demonstrates that L4.6.2.2 is responsible of RIF resistance and MDRTB progression in Nigeria". This should be further discussed.

Reviewer #2: -Are the conclusions supported by the data presented?

The results of the molecular clock phylogenetic analysis are interpreted and discussed in relation to the historical background, but the evidence is not explained at all. The authors should provide a thorough explanation, including references, to support their interpretation and discussion.

-Are the limitations of analysis clearly described?

Yes

-Do the authors discuss how these data can be helpful to advance our understanding of the topic under study?

In this paper, while the focus is primarily on presenting the analysis and findings, it also emphasizes the need for further accumulation of genomic data on tuberculosis (TB) in West Africa. By highlighting the genetic diversity, transmission patterns, and antimicrobial resistance profiles observed in the studied population, the paper implicitly suggests that expanding the genomic data repository in this region would greatly enhance our understanding of TB epidemiology and evolution.

-Is public health relevance addressed?

No

**Editorial and Data Presentation Modifications?**

Reviewer #1: Supp Mat 2, 4, 5! and 6 must be presented in a clearer way and explained further.

Table 3: remove decimal places for clarity and overview.

Reviewer #2: ＃１The authors should clearly state how they ensure data quality with respect to data downloaded from NCBI.

＃２The authors have only 24 newly read sequences in Northern Nigeria. I recommend sequencing of the 165 specimens for which drug susceptibility data were obtained.

**Summary and General Comments**

Reviewer #1: The headline should be adapted to represent better that mostly the lineages in Nigeria and the phylogenetics are focused on.

Reviewer #2: (No Response)

PLOS authors have the option to publish the peer review history of their article (what does this mean?). If published, this will include your full peer review and any attached files.

Reviewer #1: Yes: Oliver David Komm

Reviewer #2: No

Figure Files:

Data Requirements:

Reproducibility:

References

---

## [Editor Report · Decision Letter 1]

24 Aug 2023

Dear Dr. Sola,

We are pleased to inform you that your manuscript 'Mycobacterium tuberculosis complex drug-resistance, phylogenetics, and evolution in Nigeria: comparison with Ghana and Cameroon' has been provisionally accepted for publication in PLOS Neglected Tropical Diseases.

Best regards,

Simon Rayner

Academic Editor

Ana LTO Nascimento

Section Editor

---

## [Editor Report · Acceptance letter]

20 Sep 2023

Dear Pr. Sola,

We are delighted to inform you that your manuscript, "*Mycobacterium tuberculosis* complex drug-resistance, phylogenetics, and evolution in Nigeria: comparison with Ghana and Cameroon," has been formally accepted for publication in PLOS Neglected Tropical Diseases.

Best regards,

Shaden Kamhawi

co-Editor-in-Chief

Paul Brindley

co-Editor-in-Chief
